

# Development of an *in situ* Acoustic Anemometer to Measure Wind in the Stratosphere for SENSOR

Song Liang[1,2], Hu Xiong[1], Wei Feng[1], Yan Zhaoai[1,3], Xu Qingchen[1], Tu Cui[1,3]

[1]Key Laboratory of Science and Technology on Environmental Space Situation Awareness, National
Space Science Center, Chinese Academy of Sciences, Beijing 100190, China
[2]College of Earth Sciences, University of Chinese Academy of Sciences, Beijing 100049, China
[3]College of Materials Science and Opto-Electronic Technology, University of Chinese Academy of
Sciences, Beijing 100049, China

*Correspondence to*: Song Liang (songliang@nssc.ac.cn)

**Abstract**. The Stratospheric Environmental respoNses to Solar stORms (SENSOR) campaign
investigates the influence of solar storms on the stratosphere. This campaign employs a long-duration
zero-pressure balloon as a platform to carry multiple types of payloads during a series of flight
experiments in the mid-latitude stratosphere from 2019 to 2022. This article describes the development
and testing of an acoustic anemometer for obtaining *in situ* wind measurements along the balloon
trajectory. Developing this anemometer was necessary, as there is no existing commercial off-the-shelf
product, to the authors' knowledge, capable of obtaining *in situ* wind measurements on a high-altitude
balloon or other similar floating platform in the stratosphere. The anemometer is also equipped with
temperature, pressure, and humidity sensors from a Temperature-Pressure-Humidity measurement
module, inherited from a radiosonde developed for sounding balloons. The acoustic anemometer and
other sensors were used in a flight experiment of the SENSOR campaign that took place in the Da chaidan
District (95.37°E, 37.74°N) on 4 September 2019. The zonal and meridional wind speed observations,
which were obtained during level flight at an altitude exceeding 20 km, are presented. This is the first
time that *in situ* wind measurements were obtained during level flight at this altitude. In addition to wind
speed measurements, temperature, pressure, and relative humidity measurements during ascent are
compared to observations from a nearby radiosonde launched four hours earlier. Further analysis of the
wind data will presented in a subsequent publication. The problems experienced by the acoustic
anemometer during the 2019 experiment show that the acoustic anemometer must be improved for future
experiments in the SENSOR campaign.

## 1. Introduction

The response of the stratosphere to solar activities is an important scientific problem in the study of the
solar-terrestrial relationship. In theory, it is known that solar flares, proton events, and Coronal Mass
Ejections can cause sudden and global violent disturbances in the stratosphere, and that atmospheric
waves may be stimulated by short-term solar storms(Hood, 1987; Brasseur, 1993; Shindell et al., 2001;
Pudovkin, 2004; Gopalswamy et al., 2006; Labitzke, 2006; Thomas et al., 2007; Gray et al., 2010; Shi et
al., 2018). However, due to the lack of *in situ*, high-resolution, and continuous observational data in the
stratosphere, it is impossible to accurately describe how solar activities affect the mid-latitude
stratosphere. Thus, the campaign of Stratospheric Environmental respoNses to Solar stORms (SENSOR),
focusing on the above scientific research problem, has been developed (Hu, 2018). SENSOR employs a



long-duration zero-pressure balloon as the main platform to carry multiple types of payloads for conducting a series of flight experiments in the mid-latitude stratosphere from 2019 to 2022. These experiments take place during the ascending phase of solar activity. Using a high-altitude balloon as the platform has the advantages of higher temporal and spatial resolution compared to remote sensing, while enabling long-term continuous *in situ* detection. The use of a balloon also allows detection on large horizontal scales during float flight, which cannot be achieved by ground-based equipment.

An acoustic anemometer is one of the payloads carried on the high-altitude balloon to measure wind by using acoustic signals. The anemometer also contains sensors, inherited from a radiosonde, for measuring temperature, pressure, and relative humidity. The scientific goals of SENSOR are to capture and characterize the small-scale atmospheric disturbances and dynamics causing by solar activities. It is known that the enhanced ultraviolet radiation accompanying solar flares may be responsible for increases in stratospheric temperature either by increasing ozone concentrations or by increasing ozone absorption of ultraviolet radiation (Keating et al., 1987; Haigh, 1994; Hood, 2003; 2004). Particle precipitation, which mainly occurs at high latitudes, can generate large quantities of nitrogen oxides (Holt et al., 2012; Funke et al., 2014; Andersson et al., 2018) that may be transported to the mid-latitudes, reducing stratospheric ozone concentrations there and thereby causing decreases in temperature. Resulting wind disturbances, therefore, are caused by temperature changes through the thermal wind (Hood et al., 1993; Kodera, 2002; Kodera and Kuroda, 2002). An additional goal of SENSOR is to use the balloon-based anemometer to reveal small spatial and temporal characteristics of atmospheric waves in the stratosphere and to help establish a stratosphere numerical weather prediction system. The anemometer used in SENSOR can serve as a standard configuration payload on high-altitude balloons and other floating platforms to provide meteorological support in real-time.

An acoustic anemometer, which has been widely used in terrestrial environmental research due to its fast response time and high sensitivity (Kato et al., 1992; Zacharias et al., 2011; Liu et al., 2016; Bogena et al., 2018; Grachev et al., 2018; Al-Jiboori and Jaber, 2019), is a clear choice as the payload for SENSOR. However, to the authors' knowledge, there is no existing commercial off-the-shelf product that can obtain *in situ* wind measurements at the low temperatures and pressures experienced in the stratosphere. After a sonic anemometer was used in the 1950s O'Neill experiment to study turbulent fluxes in the atmospheric boundary layer (Suomi, 1957), attempts have been made to employ a similar instrument in more extreme stratospheric conditions. In the 1970s, Ovarlez et al. (1978) developed a sonic anemometer that was carried on a high-altitude balloon in 1990 to detect stratospheric fluctuations related to the Andes Mountains (de La Torre et al., 1994; de la Torre et al., 1996; Maruca et al., 2017). A 1-D anemometer, developed by Banfield et al. (2016) for Mars, was carried on a terrestrial stratospheric balloon to verify its survivability under stratospheric atmospheric conditions that are similar to those experienced on the Martian surface. Maruca et al. (2017) employed a commercial acoustic anemometer, with only modest adjustments, on a high-altitude balloon to research turbulence and obtained measurements up to an altitude of about 18 km. Despite these exceptions, the use of acoustic anemometers to study microscale meteorology has hitherto largely been limited to the troposphere (Siebert et al., 2003; Tjernström et al., 2004; Barthelmie et al., 2014; Canut et al., 2016; Maruca et al., 2017; Bodini et al., 2018; Egerer et al., 2019).

For the SENSOR campaign, we have developed an acoustic anemometer that can be used with a long-duration zero-pressure balloon, as well as on other floating platforms, such as a stratospheric airship, to employ in a series of experiments taking place from 2019 to 2022. In this article, we focus on the development of the acoustic anemometer and the preliminary experiment performed in 2019. Scientific





analyses of the measurements yielded by the anemometer will be discussed in subsequent publications. This article is arranged as follows. The principle of operation of acoustic anemometers and the development of this acoustic anemometer are introduced in Section 2. A detailed description of the 2019 balloon-borne experiments and preliminary results are presented in Section 3. Conclusions regarding the acoustic anemometer are described in Section 4.

## 2. Acoustic Anemometer

### 2.1. Principle of operation

An acoustic anemometer is used to measure wind speed by sensing the difference in propagation time of the sonic signal in the windward and leeward directions caused by the movement of airflow (Coppin and Taylor, 1983; Alberigi Quaranta et al., 1985; Fernandes et al., 2017). Taking the measurement of one-dimensional wind velocity, for example, there is a pair of transducers that are facing to each other with a distance of $L$ (as shown in Fig. 1). Each transducer can function as a transmitter as well as a receiver. To measure wind speed, each transducer transmits acoustic signals, and the opposite transducer is used as a detector to receive the signals. Due to the airflow, the flight time of sound waves in opposite directions between the pair of transducers, denoted as $t_{ww}$ and $t_{lw}$, differs. The values of $t_{ww}$ and $t_{lw}$ have the following relationships with the wind velocity in the along-transducer direction (denoted as $v$):

$$t_{ww} = \frac{L}{C-v},$$   (1)

$$t_{lw} = \frac{L}{C+v},$$   (2)

Here, $t_{ww}$ represents the travel time of signals in the windward (against the wind) direction, while $t_{lw}$ represents the travel time in the leeward (with the wind) direction. $C$ is the speed of sound. Because $t_{ww}$ and $t_{lw}$ can be directly measured, $v$ can be derived as follows:

$$v = \frac{L}{2}\left(\frac{1}{t_{lw}} - \frac{1}{t_{ww}}\right),$$   (3)

It should be mentioned that when the anemometer is on a high-altitude balloon, it measures the wind speed relative to the motion of the gondola rather than the earth-relative (absolute) wind speed. Thus, the absolute wind speed, denoted as $v_r$, is the sum of the speed obtained by the anemometer and the speed of the gondola's motion ($v_G$). If the relative wind is measured in the same direction as the gondola's motion, then $v_r$ can be expressed as:

$$v_r = v + v_G,$$   (4)

### 2.2. Instrument Design

The acoustic anemometer is one of the payloads carried by the high-altitude balloon during the SENSOR campaign. It is mainly comprised of two parts: the sensors mounted outside the balloon gondola through an aluminium alloy boom, and the electronics box installed inside the gondola. Each part is described in detail below.

The acoustic anemometer employs three pairs of ultrasonic transducers arranged in a three-dimensional structure to measure wind. The balloon flies in an environment of low temperature and low pressure, causing the acoustic signals to experience more attenuation than they would in the lower troposphere.



This attenuation increases with increasing frequency (Sutherland and Bass, 2004). To the authors' knowledge, most commercial acoustic anemometers operate at frequencies above 100 kHz, which would result in severe attenuation of acoustic signals under the conditions experienced in the stratosphere during SENSOR, possibly even making the signals indistinguishable from background noise. To improve the signal-to-noise ratio (SNR) as much as possible with a distance between transducers of 0.2 m, we have

chosen ultrasonic transducers that operate at a lower frequency of 40 kHz. This is the primary difference between the anemometer that we developed and the anemometers used in previous high-altitude balloon studies referenced in the Introduction. The received signal is amplified immediately by an ultra-low noise preamplifier to further improve the SNR, and an Automatic Gain Control (AGC) circuit is also used, different from terrestrial anemometers, to adjust its gain levels by altitude range, because the received

signal decreases as altitude increases. The different gain levels are determined by ground testing in a vacuum chamber.

    The transducers are installed on a bracket with ring structures, which are manufactured by 3-D printing to ensure that each transducer is aligned with the opposite one to maximize the SNR and to ensure that the distance between transducers are precise. The preamplifier and AGC circuits are located at the bottom

of the bracket instead of the electronic box to avoid transmission loss of the signal caused by the long cable between transducers and the electronic box.

    During flight, temperatures outside of the gondola can drop to as low as −70°C, with the lowest temperatures occurring when the balloon passes through the tropopause. Therefore, the transducers and electric devices used in the preamplifier and AGC circuits were chosen for a wide temperature range and

were tested in a thermal vacuum chamber at temperatures as low as −70°C to ensure that the transducers and circuits function under such an environment. This extreme environment has lower temperatures than the stratosphere in which the anemometer is used in SENSOR. To further protect against extreme conditions, the spaces where the circuits are placed were also been insulated.

    The amplified signals, connected to the electronic box inside of the gondola by long cables, are generated

at 1 MHz frequency by the Analog-to-Digital Converter (ADC) on a controller unit, which is one of the three boards in the electronic box. The controller board serves as a "brain" in the electrical system of the anemometer. Its core is an onboard FPGA, which operates the normal workflow of the anemometer. It generates a pulse train for the transmitting transducer, which is outputted to a Digital-to-Analog Converter (DAC), and then amplified to about 90 V (peak to peak) by the relevant driver circuits, the

second board in the electronic box. The controller board also adjusts the gain levels of the AGC circuits through a DAC according to the gondola's current altitude. Finally, the controller board employs the communication interface, RS422, with gondola's storage system; however, due to limited bandwidth, only the health data used to monitor the anemometer and a portion of the observation data can be delivered to the gondola's storage system. As the anemometer is recovered after each flight, we store the

observation data sampled by the ADC on a large-capacity storage card. This card also contains other datasets, such as the command data received by the RS422 interface from gondola's flight computer, which includes GPS time, altitude, longitude, latitude, gondola attitude, and gondola speed data.

    The third board is the power supply, which converts unregulated +28 V power provided by the gondola to regulated +12 V and +8 V power for use by the other circuits. A fuse in parallel with another fuse and

a power resister are added to the input to protect the +28V power supply in case the input current exceeds 5A. The DC/DC converter is protected by a surge protection circuit that limits the inrush current and the start-up voltage slew. An additional electromagnetic interference (EMI) filter is also used on the power lines.





In addition to wind speed, the anemometer incorporates sensors to measure temperature, pressure, and
humidity, which are located on the bracket outside of the gondola. The measurements are delivered to
the controller board through an RS232 interface on a Temperature-Pressure-Humidity measurement
module (TPH module), which was inherited from a radiosonde that we developed and used in a sounding
balloon. We retained the same circuits and sensors here. The temperature, pressure, and relative humidity
data were also stored on the large-capacity storage card.

**2.3. Data Processing**

In the current experimental design, we store the raw sampled data and perform post-processing when the
anemometer is recovered, rather than conducting real-time online processing. To obtain wind speed
following the measurement method described in Section 2.1, the acoustic signal propagation time
between transducers should be determined first. However, due to electronic delay, the acoustic signal
propagation time is not measured directly as the data is obtained. Instead, the ultrasonic waves received
by each transducer when there is no wind are obtained first and are stored in the storage card as the
reference signals. When the anemometer experiences wind, there are time differences between the
received signals and the reference signals, allowing calculation of wind speed without knowledge of the
exact electronic delay. The mathematical model employed can be illustrated as follows.
When the wind speed is zero, the measured travel time of ultrasonic waves between one pair of
transducers can be inferred as

$$t_{0r} = \frac{L}{C_0} + t_h = t_0 + t_h, \tag{4}$$

where $C_0$ is the speed of sound in the current environment, $t_h$ is the electronic delay along the
transmitted signal path, and $t_0$ is the time of flight (TOF) without electronic delay.
When the transducers are subjected to wind, the measured signal travel time in the leeward direction is

$$t_{1r} = \frac{L}{C_1 + v} + t_h = t_1 + t_h, \tag{5}$$

where $C_1$ is the speed of sound in the current environment, $v$ is the wind speed along the pair of
transducers, and $t_1$ is the TOF without electronic delay in the leeward direction. The time difference
between the received signal and the reference signal can be obtained by adopting cross correlation
between the received ultrasonic wave and the corresponding reference wave:

$$\Delta t_1 = t_{1r} - t_{0r} = t_1 - t_0, \tag{6}$$

Thus, $t_1$ would be:

$$t_1 = t_0 + \Delta t_1 = \frac{L}{C_0} + \Delta t_1, \tag{7}$$

Therefore, from Eqs. (4-7) TOF can be obtained without knowing the exact electronic delay. Similarly,
$t_2$, which is the TOF in the windward direction, can be obtained from:

$$t_2 = \frac{L}{C_0} + \Delta t_2, \tag{8}$$

The wind speed can then be obtained by substituting Eqs. (7) and (8) into Eq. (3).

As mentioned in Section 2.1, the data obtained from the anemometer are the absolute atmospheric wind
velocity relative to the motion of the gondola. Thus, with the gondola's altitude and direction provided
by the gondola's accelerometer, zonal and meridional wind speed can be obtained using Eq. (4).



### 2.4. Ground experiment

To demonstrate the functionality and reliability of our anemometer, we conducted an experiment using our anemometer and a commercial acoustic anemometer (WS500-UMB, Lufft Inc., Germany) at the summit of the Wuling Mountain (117.49°E, 40.60°N) in Xinglong County, Hebei Province, China. The
two anemometers were mounted on tripods and placed on the roof of a building at the top of the mountain. This location has an open view; obstructions are much lower than the tripods' height and at a substantial distance from the tripods (Fig. 3). Thus, the natural airflow was little-disturbed and the two anemometers were approximately in the same airflow field. To compare the anemometers' measurements with greater temporal precision, GPS was used to synchronize the time measurements of the two anemometers.
The measurement repetition rate of the commercial anemometer was 0.1 Hz while that of our own anemometer was 1 Hz; thus, the minimum time interval for data comparison was 10 s. To reduce the error between measurements caused by the different sampling times as much as possible, data from each anemometer were averaged in 1-minute intervals before comparison.

According to the measurements from both anemometers shown in Fig. 4a, the wind speed measured by
our anemometer agrees well with the commercial anemometer's results. The wind speed at the top of Wuling Mountain ranges from 1 m s$^{-1}$ to 3 m s$^{-1}$ during the observation period with an average of approximately 1.8 m s$^{-1}$. Wind speed differences, which are calculated by subtracting our anemometer's measurements from the commercial anemometer's measurements, reach a maximum absolute value of 0.9 m s$^{-1}$ during the evaluation period (Fig. 4b). The mean wind speed difference is −0.0157 m s$^{-1}$, with
a standard deviation of 0.3016 m s$^{-1}$. Overall, this test shows that our anemometer may have a slight high bias relative to the commercial anemometer. However, the differences between the two anemometers may be due to different conditions experienced by the two instruments, as there is no guarantee that two anemometers measure the same microscale air mass, no matter their proximity. Overall, the similarity of the results between our anemometer and the commercial anemometer allows us to conclude that our
anemometer is capable of accurately measuring wind speed and that it generates output that is reasonable and in line with our expectations.

### 3. Results

The flight experiments of the SENSOR campaign during 2019 took place in the Da chaidan District (95.37°E, 37.74°N), Qinghai Province, China. Fig. 5 shows the photograph of the gondola with all
payloads assembled before launching at the experiment site.

The balloon was launched at approximately 16:18 UTC on 4 September. The flight trajectory is shown in Fig. 6. The balloon reached its maximum altitude of approximately 25 km at 17:56 UTC; level flight lasted for approximately 1.5 hours until 19:42 UTC when the gondola was separated from the balloon by a cutter. The anemometer was powered on at the start of the experiment and functioned until the end of
the float flight. When the gondola landed, the anemometer was recovered successfully; wind, temperature, pressure, and relative humidity measurements were obtained from the flight.

In this experiment, to acquire the best SNR and avoid damaging the transducers, we had set the driver voltages as high as possible and adjusted the AGC circuit's gain levels automatically as the balloon rose. Although everything worked normally in the ground environment, during the balloon experiment the
output signals from the ultrasonic transducers unexpectedly overflowed the ADC's input voltage range after launching. The signals did not return to normal until the balloon rose above an altitude of 10 km. Additionally, large spikes in the wind speed measurements were observed, the same phenomenon as



Maruca et al. (2017) reported. To the best of the authors' knowledge, these spikes were caused by the large attenuation of acoustic signals under low pressure, which led to the misjudgement of propagation time between transducers. However, it was straightforward to eliminate these spurious wind speed spikes, as they were clearly distinct from the data immediately preceding and following them.

Figs. 7 and 8 show a 300-s period of zonal and meridional wind speed measurements, with a repetition rate of 1 Hz, after the balloon began level flight at 17:56:27 UTC. The top panels of these figures show that the absolute values of the zonal and meridional wind measured by the anemometer never exceeded 1 m s$^{-1}$. Thus, the movement of the gondola largely followed the air flow. However, the middle panels of Figs. 7 and 8 also demonstrate that the wind speed measurements (blue line) from the anemometer, which combine relative wind velocity and the gondola movement speed (red line), were more sensitive to changes in wind than the gondola movement speed was. The principal reason is that the gondola had a large volume and a mass exceeding 400 kg; thus, its zonal and meridional speeds did not change immediately from disturbances in the wind. The results from this experiment demonstrate that the acoustic anemometer employed in this study can sense rapid changes in wind and is useful for researching microscale wind fluctuations in the stratosphere. Though the instrument experienced problems during this experiment, this was the first time, to the authors' knowledge, that wind measurements during level flight above an altitude of 20 km were obtained by an acoustic anemometer. The results presented here also show that the anemometer can be further improved to perform more reliably in future experiments.

The temperature, pressure, and relative humidity experienced by the gondola were also measured with a repetition rate of 1 Hz by their respective sensors on the anemometer. These data were compared with measurements from a radiosonde launched at 12:00 UTC, approximately four hours before the experiment, from Golmud Observation Station (94.90°E, 36.42°N), as shown in Fig. 9. The radiosonde data were downloaded from the University of Wyoming website. Although the vertical resolution of the radiosonde data was much coarser than our measurements, they still could be used to verify our measurements. Comparing the results shows that the temperature and pressure measurements had good consistency, while relative humidity had the same trend in our measurements and the radiosonde measurements. The main reason for relative humidity differences may be that the two experiments were conducted four hours apart in different locations. It is also possible that one or both humidity sensors experienced systematic errors. Besides, from plot (9a), it is clear that the tropopause height over the Qinghai Tibet Plateau was about 17 km during both sets of measurements, and tropopause temperatures reached values below −70°C.

### 4. Conclusions

An acoustic anemometer has been developed for the SENSOR campaign, which is carried aboard a high-altitude balloon to measure wind velocity. This acoustic anemometer obtains zonal and meridional wind speed measurements using the principle that the propagation time of ultrasonic signals differs between the leeward and windward directions. Additionally, the anemometer contains a radiosonde-based module that obtains temperature, pressure, and relative humidity measurements during flight.

The anemometer participated in a flight experiment of the SENSOR campaign in 2019, which took place in the Da chaidan District (95.37°E, 37.74°N). During this experiment, the anemometer obtained continuous wind velocity data above an altitude of 20 km. This is the first time, to the authors' knowledge, that an acoustic anemometer has obtained wind velocity data during level flight at that altitude. Results from our experiment provide confidence in using this technology to measure wind in the stratosphere,





although much work remains necessary to improve the performance of the instrument. Temperature, pressure, and relative humidity data are also obtained from the time of launch to the end of level flight. Observations from the ascent phase were compared with the radiosonde data from the Golmud Observation station (94.90°E, 36.42°N) approximately four hours earlier, and the results show very good agreement. A scientific analysis of the observed data from this flight is currently underway and will be

presented in detail in a future publication.

Three more high-altitude balloon experiments in the SENSOR campaign will be conducted during 2020, 2021 and 2022. The acoustic anemometer described in this article will serve as a standard configuration payload during these experiments. Despite the success of this experiment, the acoustic anemometer must be improved to obtain more reliable measurements of short-term and microscale fluctuations in

stratospheric wind during periods of elevated solar activity, a key scientific objective of the SENSOR campaign.

*Data availability*. This article focuses on the development of acoustic anemometer. Observation data from the flight experiment are not currently publicly available since the further analysis of them is under

way.

*Author Contributions*. Conceptualization and methodology, SL and HX; Writing—original draft preparation, SL; Writing—review and editing, SL, HX, WF, YZ, XQ and TC; All authors have read and agreed to the published version of the manuscript.


*Competing interests*. The authors declare that they have no conflict of interest.

*Acknowledgments*. This work is supported by the Strategic Priority Research Program of the Chinese Academy of Sciences (Grant Nos. XDA17010303, XDA17010302, XDA17010301). We gratefully

acknowledge Aerospace Information Research Institute, Chinese Academy of Sciences, for providing and successfully launching the high altitude balloon. We would like to express our gratitude to University of Wyoming website for providing the radiosonde data for comparison. We thank LetPub (www.letpub.com) for its linguistic assistance and scientific consultation during the preparation of this manuscript.

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



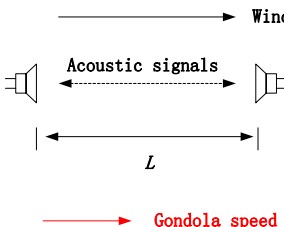

Figure 1 Diagram of the principle of measuring wind speed in a single direction using one pair of transducers.




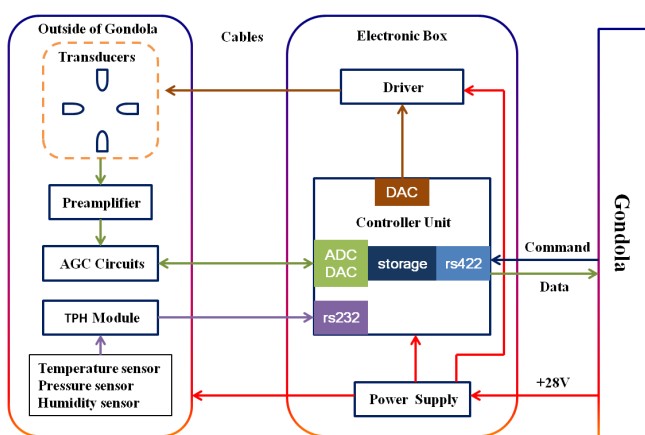

Figure 2 Block diagram of the electrical system associated with the acoustic anemometer.




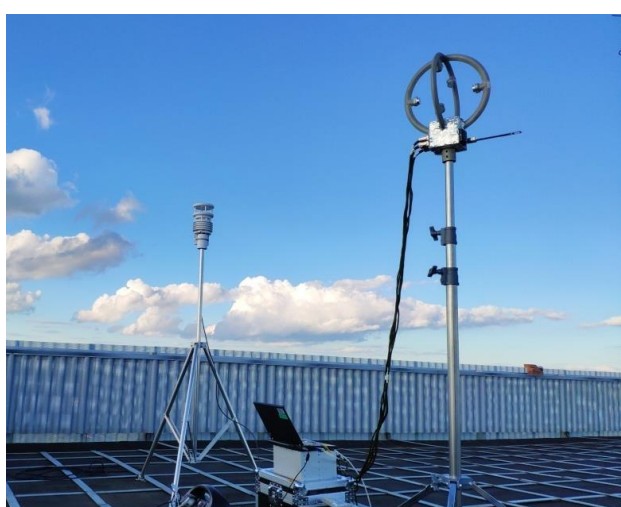

Figure 3 Photograph of the acoustic anemometer developed in this study alongside a commercial anemometer at the summit of the Wuling Mountain.






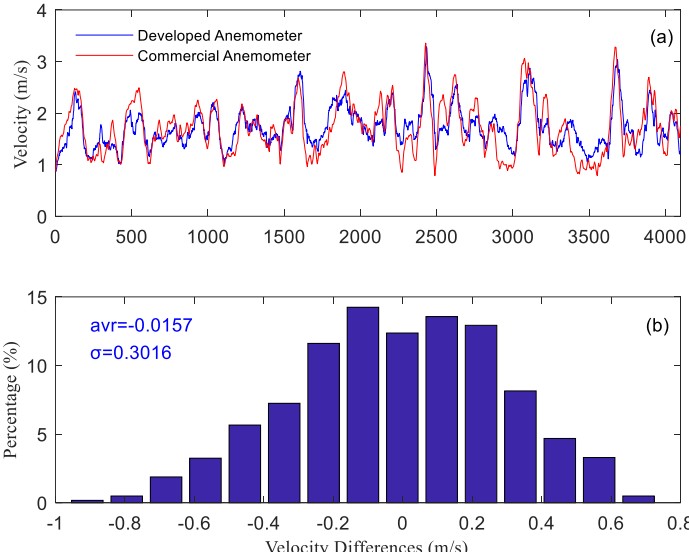

Figure 4 Comparison of observations at the top of Wuling Mountain between the anemometer developed in this study and the commercial anemometer. In (a), the blue line represents data obtained from the anemometer developed in this study while the red line represents data obtained from the commercial anemometer. In (b), the distribution of the differences between the commercial anemometer and our anemometer are shown. All data in this figure are averaged in 1-minute intervals to minimize error caused by the different sampling times of the two anemometers.




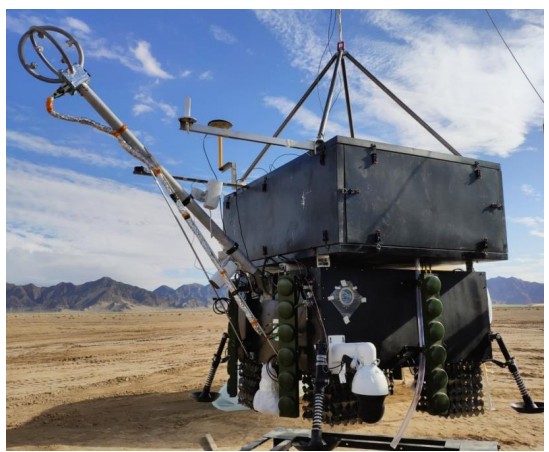

Figure 5 Photograph of the gondola with payloads prior to ascent on 4 September 2019.





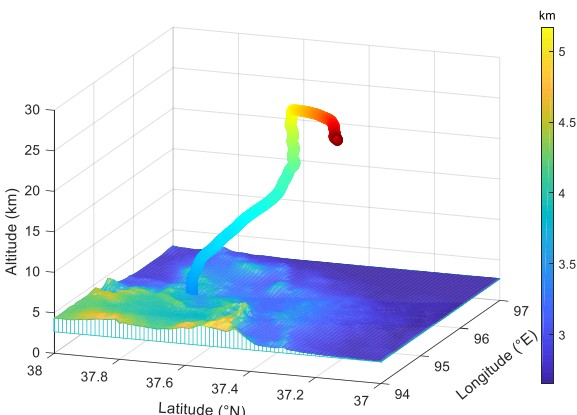

Figure 6 Flight trajectory (color line, the dark red stands for the end of the float flight) of the gondola
during the 4 September 2019 experiment. The colorbar represents the topography of the experiment site.

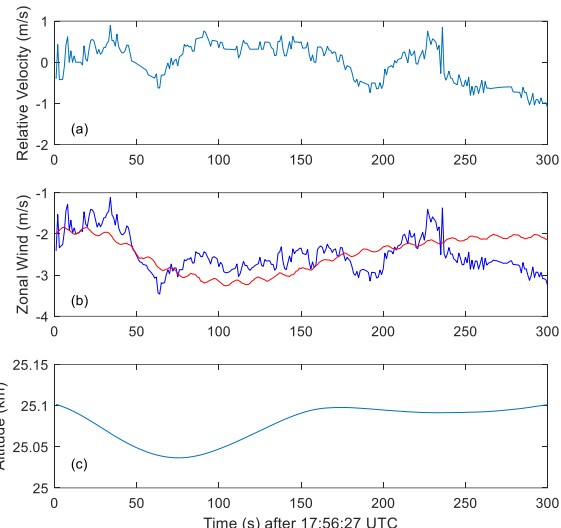

Figure 7 For a 300-s period at the commencement of level flight, starting at 17:56:27 UTC: (a) relative
zonal wind speed measured by the anemometer, (b) comparison between relative zonal wind speed (blue)
and gondola zonal movement speed (red), and (c) gondola altitude.



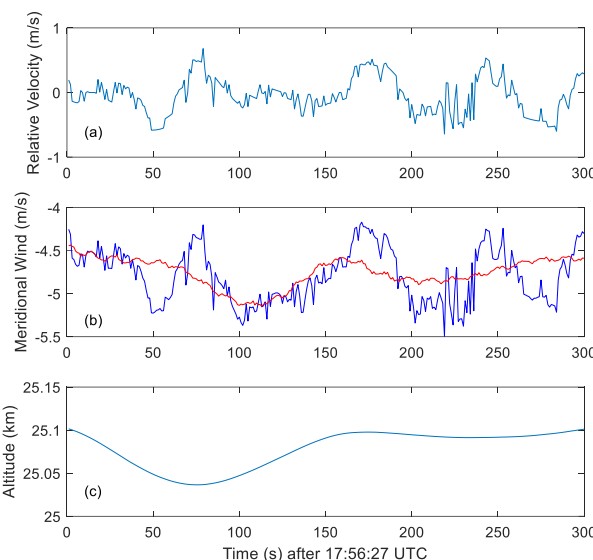

Figure 8 As in Fig.7, but for the meridional wind.

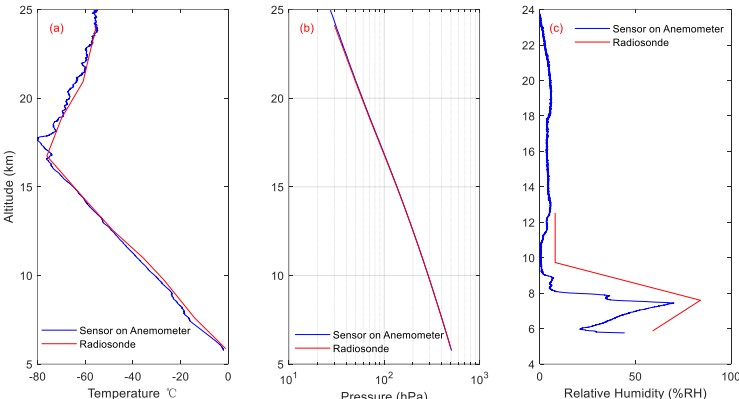

Figure 9 Comparisons of (a) temperature, (b) pressure, and (c) relative humidity between measurements
from sensors on the anemometer and radiosonde data obtained from the Golmud Observation Station
approximately four hours earlier.