# Peer review of "Development of an *in situ* Acoustic Anemometer to Measure Wind in the Stratosphere for SENSOR"

_Atmospheric Measurement Techniques, 2021_

## Referee Comment (RC1)

Review about
**"Development of an in situ Acoustic Anemometer to Measure Wind in the Stratosphere for SENSOR"**
by
Song Liang, Hu Xiong, Wei Feng, Yan Zhaoai, Xu Qingchen, and Tu Cui

The manuscript describes the development of a homemade ultrasonic anemometer for use on a drifting balloon for stratospheric wind observations. Ground-based intercomparisons with a commercial available ultrasonic anemometer are presented, followed by one flight example with a stratosphere balloon drifting in about 20 km height. First of all congratulations to this general success, I have a rough sense of the technical requirements to develop such a device and get it airborne. However, at several points in this manuscript I have the feeling that I read more a technical progress report instead of a publication for a broader audience. Within the introduction I would expect a deeper discussion of what kind of environmental conditions I have to deal with and why is a device of the shell not suitable? Furthermore, what are specific scientific questions you want to answer with the new device? This would immediately set the bounds for the desired specifications for a new device. Historically, ultrasonic anemometers have been developed to resolve atmospheric turbulence and state-of-the-art sensors have time response of 10 ms or so – it might be due to the low pressure conditions that your ultrasonic is limited by 1 Hz resolution but this issue has to be discussed in detail. Furthermore, a deeper discussion of the principle problems of measuring with ultrasonic anemometers under stratospheric conditions should be the main motivation. In the current state I cannot suggest to consider this manuscript to be published in *Atmospheric Measurement Technology.* I will provide more detailed comments and suggestions for a revised version below.

Detailed Comments and Suggestions:

1. Abstract: At the end of the abstract you mention, "*Further analysis of the wind data will be presented in a subsequent paper".* I immediately ask myself why not presenting more analysis in this paper? The analysis here is somewhat superficial and even if you want to convince the reader that you have developed a nice instrument this can be done best by a more detailed data analysis. Furthermore, if you realize that the sensor needs improvements so why wait until you have solved all issues before publish the results?

2. Introduction line 43ff: a drifting balloon definitively provides useful insight in the structure of the stratosphere and has several advantages but also disadvantages compared to remote sensing such as problems with instationarity – this should be discussed here.

3. Introduction (general): the structure of this paragraph should be improved; you jump a little bit between motivation, technical requirements and scientific goals.

4. Introduction line 56ff: You explicitly mention that the anemometer should be designed for small-scale observations in the stratosphere, please specify the requirements for the new device!

5. Introduction line 65ff: You mention the extreme environment in which the device should work: please specify!

6.  Introduction line 68ff: you cite the work of Ovarlez et al., 1978: Why is this ultrasonic anemometer not suitable for your application? Or what want you doing better compared to their device?

7.  Section 2.1: The introduction provides a lot of material you can read in most textbooks about ultrasonic anemometers, the comment on line 106 about the measurement of wind speed in a moving reference system is a general challenge with airborne wind measurements. Please provide at least one citation. However, with a more or less tracer-like floating balloon this is an even more interesting problem because the balloon motion is quasi-Lagrangian – this should be discussed in much more detail as it introduce an interesting point.

8.  Section 2.1: There is no discussion at this point how $v_g$ is measured on the balloon

9.  Section 2.2 line 119 ff: Here you mention the first time which kind of technical problems and challenges you have with the anemometer under low pressure and temperature conditions. I think this is the key motivation for developing such a system – otherwise you could use a device off the shell – right? This information should be – with much more details – presented at a more prominent place in the manuscript!

10. Line 137: here you mention the first time typical temperature conditions for your device – I suggest putting it much earlier.

11. Section 2.2 includes in general a lot of technical – and partly trivial - details that apply for all sensors such as the information that you used a fuse to protect you system. I am more interested in aspects like what is different to other acoustic anemometers and how did you solved problems related to the extreme environment of the stratosphere.

12. Section 2.3 "Data processing": Similar comment as above: the information you have provided here applies to all ultrasonic anemometers and is generally valid, but not specifically for your device - at least I don't see any information that refers to the problems caused by low temperature or pressure.

13. Section 2.4: Ground Experiment: The argumentation that both sensors experience the same airflow is somewhat vague; I think one can learn something about the sensors but here it is the first time you mention the temporal resolution of you device which I consider as remarkable low for an ultrasonic anemometer, however, the "reference" system is even slower with a resolution of 10 s? Usually, ultrasoncis are considered as turbulence sensors with sampling frequencies of up to 100 Hz – what are your technical limitations? It might be that your device is slower to meet the conditions of low pressure but you should explain it?

14. A technical question: In Fig 5 it seems that the mechanical structure is quite robust but have you considered effects such as flow distortions or transducer shadowing effects which are a key issue for sonics? – Maybe with a lower resolution this is not an issue but you should at least consider such hazards.

15. Section 2.4 line 216ff: is the observed wind speed the range you expect as relative airspeed for your balloon experiment? Why do you provide four digits for mean wind speeds? What is the useful resolution (absolute accuracy) of the both sonics?

16. Section 2.4: I am not sure what I can learn from this intercomparison experiment but this is partly also due to the fact that I don't know what you expect from your new device.

17. Section 3 "Results": I have serious problems with interpreting the wind measurements at all – despite any technical problems such as the observed spikes: I assume even a zero-pressure balloon with a mass of 400 kg (gondola) does not exactly moves with the mean flow but what are the differences? You measure a relative flow speed but how should I interpret the red line in Fig 7b? By the way, here you only show the zonal wind vector component but you mentioned earlier that you have developed a three-dimensional wind measurement device – right? What about the other components and how is the 3d-motion of the balloon be measured? Are you determine also the angular rates and attitude angles?

18. If you have a radiosonde available for comparison: why not compare wind speeds? From my point of view, this would be the most obvious thing to do and one could discuss the topic of the quasi-Lagrangian observation in more detail.

---

## Author Comment (AC1)

**Responses from authors to Reviewer 2's comments**

**Ref. No.: amt-2021-76**

**Title: Development of an in situ Acoustic Anemometer to Measure Wind in the Stratosphere for SENSOR**

Song Liang1,2, Hu Xiong1, Wei Feng1, Yan Zhaoai1,3, Xu Qingchen1, Tu Cui1,3 1Key Laboratory of Science and Technology on Environmental Space Situation Awareness, National Space Science Center, Chinese Academy of Sciences, Beijing 100190, China

2College of Earth Sciences, University of Chinese Academy of Sciences, Beijing 100049, China

3College of Materials Science and Opto-Electronic Technology, University of Chinese Academy of Sciences, Beijing 100049, China

**Reviewer 2's comments:**

The scope of this manuscript is to present the development and the performance of a sonic anemometer able to produce wind measurements in the stratosphere. While I think sensors technology is now mature to design experiments based on ultrasonic probes purposely developed for high altitude atmospheric observations, I have major concerns on the quality and the originality of the research proposed here.

Although the authors cite two recent articles presenting experiments that fully achieved the goal of performing science quality measurements in the stratosphere with acoustic anemometers, they do not reference them properly and instead they make such statements as "This is the first time that in-situ wind measurements were obtained during level flight at this altitude" (meaning above 20km), as reported in the abstract at lines 22-23.

This is misleading, as I will explain in the following, and the authors insist throughout the text on the fact that their measurements are (the first) being performed above 20km and during a balloon level flight, in order to differentiate their work from previous experiments based on this technology already performed in the stratosphere.

As a matter of fact, Banfield et al. 2016 and Maruca et al. 2017 (both cited in the maniscript) performed experiments in which sonic anemometers have been developed (and/or modified) and tested with positive outcomes on high altitude stratospheric balloons. In the case of Banfield et al. 2016 the probe operated up to ~ 33 km while the sonic anemometer of the TILDAE experiment by Maruca et al. 2017 operated up to around 19 km. These experiments (dated back in 2015 and 2016, respectively) have been successful attempts of employing sonic anemometers for stratospheric measurements and they both returned science quality data, as testified by the statistical analyses presented in the aforementioned manuscripts, including the computation of kinetic energy spectra (see Maruca et al. 2017).

Indeed, what is relevant for these type of the experiments is not the peak altitude at which a sonic anemometer returned some sort of signal, but the fact that ultrasonic probes have been able to produce reliable measurements in the stratosphere - meaning above the tropopause - and that these measurements could be used to perform rigorous scientific investigations. These goals have not been achieved by the experiment presented here, since the signals reported in the plots included in the manuscript clearly show that the probe needs further development and testing, and no analysis of the data

collected has been performed.

On the sidebar, I would like to point out that the tropopause does not have the same altitude everywhere over the globe and it is lower at the poles, where the ultrasonic probe by Maruca et al. 2017 was operated. Thus the maximum operational altitude of 19 km reported in Maruca et al. 2017 is probably deeper in the stratosphere than the altitude of 20 km over the Da chaidan district (as reported in the present manuscript). Even the evidence that the probe presented here has been tested during a level flight is rather weak, since Fig.7 shows a time series of only 300 seconds during which the altitude of the balloon was more or less constant. This time interval is really too short. However, following the narrative of the manuscript, this point should differentiate significantly the present work from Banfield et al. 2016 and Maruca et al. 2017, where ultrasonic anemometers operated only during the ascent phase of the the respective balloon flights.

For these reasons I cannot suggest the publication of this manuscript on AMT. Though, I strongly encourage the authors to pursue with the development of their acoustic anemometer and to re-propose this work corroborated by the analysis of the data collected, once its design will allow to perform science valuable measurements in the stratosphere.

**Answers to the Reviewer 2's comments:**

Thank you very much for your time and efforts reviewing this study. The answers that we have made based on the reviewer's comments are discussed below "point-by-point". Please kindly find the following responses (the comments are shown in italics and blue while answers in non-italics and red).

**Q1:**

The scope of this manuscript is to present the development and the performance of a sonic anemometer able to produce wind measurements in the stratosphere. While I think sensors technology is now mature to design experiments based on ultrasonic probes purposely developed for high altitude atmospheric observations, I have major concerns on the quality and the originality of the research proposed here.

Although the authors cite two recent articles presenting experiments that fully achieved the goal of performing science quality measurements in the stratosphere with acoustic anemometers, they do not reference them properly and instead they make such statements as "This is the first time that in-situ wind measurements were obtained during level flight at this altitude" (meaning above 20km), as reported in the abstract at lines 22-23.

This is misleading, as I will explain in the following, and the authors insist throughout the text on the fact that their measurements are (the first) being performed above 20km and during a balloon level flight, in order to differentiate their work from previous experiments based on this technology already performed in the stratosphere.

As a matter of fact, Banfield et al. 2016 and Maruca et al. 2017 (both cited in the maniscript) performed experiments in which sonic anemometers have been developed (and/or modified) and tested with positive outcomes on high altitude stratospheric balloons. In the case of Banfield et al. 2016 the probe operated up to  $\sim 33$  km while the sonic anemometer of the TILDAE experiment by Maruca et al. 2017 operated up to around 19 km. These experiments (dated back in 2015 and 2016, respectively) have been successful attempts of employing sonic anemometers for stratospheric

measurements and they both returned science quality data, as testified by the statistical analyses presented in the aforementioned manuscripts, including the computation of kinetic energy spectra (see Maruca et al. 2017).

Thank you for your comment. Banfield et al. 2016 and Maruca et al. 2017 did a good job in developing and testing sonic anemometers on high altitude balloons. The focus of our work is, based on drawing experiences from their work, to take further improvements to the acoustic anemometer according to the atmospheric environment at the float flight altitude (~25km) of the balloon we used. Our anemometer had been tested in the experiment and obtained measurements during float flight. These are the major contribution of our work. A preliminary analysis of the data was also added according to your suggestions (please see answers to Q2).

In the TILDAE experiment by Maruca et al. 2017, the modified sonic anemometer can operate up to around 19km, above that altitude only fill values were returned due to "Almost assuredly, this was the result of the decrease in atmospheric pressure during the ascent". As to the balloon test by Banfield et al. 2016, they didn't show the wind measurements obtained from the sonic anemometer.

Above all, we had drawn the conclusion "This is the first time that in-situ wind measurements were obtained during level flight at this altitude".

We have added the following discussion on what efforts we had taken to accommodate our acoustic anemometer to the high-altitude atmosphere in our revised manuscript.

In our experiment, the high-altitude balloon were drifting at the altitude of about 25km, where the atmosphere had significant difference from terrestrial environment: low pressure of about 30hPa and low temperature with extremes approaching  $\sim$ -70 °C during the balloon's ascent. In order to make sure our anemometer can operate at such an altitude, the characteristics of acoustic signals propagation attenuation in the atmosphere had been analyzed.

According to *Bass et al., 1990*, Bass et al., 1995 and *Sutherland and Bass, 2004*, when the sound wave propagates in the atmosphere, the signal attenuation caused by atmospheric absorption is mainly related to the acoustic frequency and atmospheric pressure, attenuation coefficients  $\alpha$  in dB per meters (dB/m) can be expressed as follows:

$$\alpha = 8.686f^{2} \{ 1.84 \times 10^{-11} \left(\frac{p}{p_{0}}\right)^{-1} \left(\frac{T}{T_{0}}\right)^{1/2} + \left(\frac{T}{T_{0}}\right)^{-5/2} \times \left[ 0.01278 \frac{e^{-2239.1/T}}{f_{r,o} + f^{2}/f_{r,o}} + 0.1068 \frac{e^{-3352/T}}{f_{r,N} + f^{2}/f_{r,N}} \right] \}$$

Where f is the acoustic frequency in Hz, p is the atmospheric pressure in Pa,  $p_0$  is the reference atmospheric pressure in Pa, T is the atmospheric temperature in K,  $T_0 = 293.15K$ , is the reference atmospheric temperature,  $f_{r,o}$ ,  $f_{r,N}$  are the relaxation frequency of molecular oxygen and the relaxation frequency of molecular nitrogen, respectively:

$$f_{r,o} = \frac{p}{p_0} \Big( 24 + 4.04 \times 10^4 h \frac{0.02 + h}{0.391 + h} \Big),$$

$$f_{r,N} = \frac{p}{p_0} \left(\frac{T_0}{T}\right)^{1/2} \left(9 + 280h \exp\{-4.17\left[\frac{T_0}{T}\right]^{1/3} - 1\right]\right),$$

Where h is the molar concentration of water vapor in percent.

According to the above formulas, figure 1 shows the variation of different frequencies of acoustic signals attenuation caused by atmospheric absorption with height at a distance of 0.2m from the acoustic source.

Figure 1 Atmospheric absorption attenuation of different frequencies of acoustic signals.

The atmospheric absorption attenuation of acoustic signal increases with the increase of acoustic frequency and with the decrease of atmospheric pressure that goes down exponentially with height. At the balloon level flight altitude of about 25km, the received signal intensity with acoustic frequency of 40kHz is at least 10dB higher than that of signals with frequencies of above 100kHz. Therefore, in our acoustic anemometer, the sensors with resonant frequency of 40kHz had been used to achieve higher Signal-to-Noise Ratio (SNR), which is the primary difference between the anemometer that we developed and the anemometers used by Banfield et al. 2016 and Maruca et al. 2017. Besides, an Automatic Gain Control (AGC) circuit is also used, different from terrestrial anemometers, to adjust its gain levels with altitude range to obtain better SNR.

In addition, to avoid the flow distortion from the gondola, the sensor bracket was not mounted outside of the gondola directly, but was through a boom with a length of 1.8m and an elevation angle of 45°. According to Lenoir et al., 2011, the perturbation from the gondola has little influence on the measurements.

**Q2:**

Indeed, what is relevant for these type of the experiments is not the peak altitude at which a sonic anemometer returned some sort of signal, but the fact that ultrasonic probes have been able to produce reliable measurements in the stratosphere - meaning above the tropopause - and that these measurements could be used to perform rigorous scientific investigations. These goals have not been achieved by the experiment presented here, since the signals reported in the plots included in the manuscript clearly show that the probe needs further development and testing, and no analysis of the data collected has been performed.

Thank you for your comment. We added a preliminary analysis of the data in the revised manuscript as follows.

The internal data sampling rate of the acoustic anemometer we developed is 10Hz. In order to improve the signal-to-noise ratio, the original sampled signal within 1s are accumulated, thus the data update rate we given in the manuscript is 1Hz (as shown in Fig. 7 and Fig. 8 in the manuscript). Here, we show the measurements above an altitude of 21km with update rates of 10Hz and 1Hz, respectively.

Figure 2 For a 1900-s period starting at 17:33:07 UTC: (a) relative zonal wind speed measured by the anemometer at update rates of 10Hz (cyan) and 1Hz (red), respectively, (b) comparison between zonal wind speed (red) and gondola zonal movement speed (blue), and (c) gondola altitude.

---

## Author Comment (AC2)

**Responses from authors to Reviewer 1's comments**

**Ref. No.: amt-2021-76**

**Title: Development of an in situ Acoustic Anemometer to Measure Wind in the Stratosphere for SENSOR**

Song Liang[1,2], Hu Xiong[1], Wei Feng[1], Yan Zhaoai[1,3], Xu Qingchen[1], Tu Cui[1,3]

[1]Key Laboratory of Science and Technology on Environmental Space Situation Awareness, National Space Science Center, Chinese Academy of Sciences, Beijing 100190, China

[2]College of Earth Sciences, University of Chinese Academy of Sciences, Beijing 100049, China

[3]College of Materials Science and Opto-Electronic Technology, University of Chinese Academy of Sciences, Beijing 100049, China

*Reviewer 1's comments:*

*1- Abstract:At the end of the abstract you mention, "Further analysis of the wind data will be presented in a subsequent paper". I immediately ask myself why not presenting more analysis in this paper? The analysis here is somewhat superficial and even if you want to convince the reader that you have developed a nice instrument this can be done best by a more detailed data analysis. Furthermore, if you realize that the sensor needs improvements so why wait until you have solved all issues before publish the results?.*

*2- Introduction line 43ff: a drifting balloon definitively provides useful insight in the structure of the stratosphere and has several advantages but also disadvantages compared to remote sensing such as problems with instationarity - this should be discussed here.*

*3- Introduction (general): the structure of this paragraph should be improved; you jump a little bit between motivation, technical requirements and scientific goals.*

*4- Introduction line 56ff: You explicitly mention that the anemometer should be designed for small- -scale observations in the stratosphere, please specify the requirements for the new device.*

*5- Introduction line 65ff: You mention the extreme environment in which the device should work: please specify.*

*6- Introduction line 68ff: you cite the work of Ovarlez et al., 1978: Why is this ultrasonic anemometer not suitable for your application? Or what want you doing better compared to their device?*

*7- Section 2.1: The introduction provides a lot of material you can read in most textbooks about ultrasonic anemometers, the comment on line 106 about the measurement of wind speed in a moving reference system is a general challenge with airborne wind measurements. Please provide at least one citation. However, with a more or less tracer- - like floating balloon this is an even more interesting problem because the balloon motion is quasi- -Lagrangian- this should be discussed in much more detail as it introduce an interesting point.*

*8- Section 2.1: There is no discussion at this point how vg is measured on the balloon.*

*9- Section 2.2 line 119 ff: Here you mention the first time which kind of technical problems and challenges you have with the anemometer under low pressure and temperature conditions. I think this is the key motivation for developing such a system – otherwise you could use a device off the shell – right? This information should be – with much more details – presented at a more prominent place in the manuscript!*

*10- Line 137: here you mention the first time typical temperature conditions for your device – I suggest putting it much earlier.*

11- Section 2.2 includes in general a lot of technical – and partly trivial - - details that apply for all sensors such as the information that you used a fuse to protect you system. I am more interested in aspects like what is different to other acoustic anemometers and how did you solved problems related to the extreme environment of the stratosphere.

12- Section 2.3 "Data processing": Similar comment as above: the information you have provided here applies to all ultrasonic anemometers and is generally valid, but not specifically for your device - - at least I don't see any information that refers to the problems caused by low temperature or pressure.

13- Section 2.4: Ground Experiment: The argumentation that both sensors experience the same airflow is somewhat vague; I think one can learn something about the sensors but here it is the first time you mention the temporal resolution of you device which I consider as remarkable low for an ultrasonic anemometer, however, the "reference" system is even slower with a resolution of 10 s? Usually, ultrasoncis are considered as turbulence sensors with sampling frequencies of up to 100 Hz – what are your technical limitations? It might be that your device is slower to meet the conditions of low pressure but you should explain it?

14- A technical question: In Fig 5 it seems that the mechanical structure is quite robust but have you considered effects such as flow distortions or transducer shadowing effects which are a key issue for sonics? – Maybe with a lower resolution this is not an issue but you should at least consider such hazards.

15- Section 2.4 line 216ff: is the observed wind speed the range you expect as relative airspeed for your balloon experiment? Why do you provide four digits for mean wind speeds? What is the useful resolution (absolute accuracy) of the both sonics?

16- Section 2.4: I am not sure what I can learn from this intercomparison experiment but this is partly also due to the fact that I don't know what you expect from your new device.

17- Section 3 "Results": I have serious problems with interpreting the wind measurements at all – despite any technical problems such as the observed spikes: I assume even a zero- - pressure balloon with a mass of 400 kg (gondola) does not exactly moves with the mean flow but what are the differences? You measure a relative flow speed but how should I interpret the red line in Fig 7b? By the way, here you only show the zonal wind vector component but you mentioned earlier that you have developed a three- - dimensional wind measurement device – right? What about the other components and how is the 3d- - motion of the balloon be measured? Are you determine also the angular rates and attitude angles?

18- If you have a radiosonde available for comparison: why not compare wind speeds? From my point of view, this would be the most obvious thing to do and one could discuss the topic of the quasi- - Lagrangian observation in more detail.

**Answers to the Reviewer1's comments:**
**Thank you very much for your time and efforts reviewing this study. The answers that we have made based on the reviewer's comments are discussed below "point-by-point". Please kindly find the following responses (the comments are shown in italics and blue while answers in non-italics and red).**

*1- Abstract:At the end of the abstract you mention, "Further analysis of the wind data will be presented in a subsequent paper". I immediately ask myself why not presenting more analysis in this paper? The analysis here is somewhat superficial and even if you want to convince the reader that you have developed a nice instrument this can be done best by a more detailed data analysis. Furthermore, if you realize that the sensor needs improvements so why wait until you have solved all issues before publish the results?.*

**Thank you for your comment and suggestion. A preliminary analysis of the data was added in our**

**revised manuscript as follows according to your suggestion.**

The data update rate we given in the manuscript is 1Hz (as shown in Fig. 7 and Fig. 8 in the manuscript). In fact, the internal sampling rate of the acoustic anemometer we developed is 10Hz. For improving the signal-to-noise ratio (SNR), the original sampled signal within 1s are accumulated. Here, with regard to a preliminary spectral analysis of the data, we show the measurements above an altitude of 21km with update rates of 10Hz and 1Hz, respectively.

[Figure]

Figure 1 For a 1900-s period starting at 17:33:07 UTC: (a) relative zonal wind speed measured by the anemometer at update rates of 10Hz (cyan) and 1Hz (red), respectively, (b) comparison between zonal wind speed (red) and gondola zonal movement speed (blue), and (c) gondola altitude.

[Figure]

Figure 2 As in Fig.1, but for the meridional wind.

[Figure]

Figure 3 (a) vertical wind speed measured by the anemometer at update rates of 10Hz (cyan) and 1Hz (red), respectively, and (b) gondola altitude.

The power spectral density of the 10Hz measurements during the float flight period (from 1500s to 1900s) is evaluated using periodogram method. (as shown in figure 4).

[Figure]

Figure 4 Spectral analysis of the 10Hz measurements during the float flight period (from 1500s to 1900s): (a) relative horizonal wind speed, and (b) verical speed. The red line in each case indicates the theoretical spectral trend of -5/3.

According to Kolmogoroff theory (Kolmogorov, 1941), turbulence spectra can be well accepted to have specific gradient characteristics: a -5/3 slope in the inertial sub-range. From the measured spectral shown in figure 3, a clear -5/3 slope decay at frequencies from 0.02 to 0.3Hz indicates that there were turbulence exist at the float flight altitude.

*2- Introduction line 43ff: a drifting balloon definitively provides useful insight in the structure of the stratosphere and has several advantages but also disadvantages compared to remote sensing such as problems with instationarity - this should be discussed here.*

Thank you for your suggestion. The main shortcomings of a drifting balloon are its short flight time and the altitude instability at floating area. In SENSOR campaign, the balloon had the ability to fly tens of hours or even longer. This had met the scientific research needs. As to the altitude instability, balloon neutral oscillations were very slow, the period of which was usually in a few minutes. The influence on the measurements of payloads could be ignored or be removed in data analysis.

*3- Introduction (general): the structure of this paragraph should be improved; you jump a little bit between motivation, technical requirements and scientific goals.*

**Thank you for your suggestion. The structure of this paragraph have been improved. Please kindly find the following revision.**

**Before revision:**
The response of the stratosphere to solar activities is an important scientific problem in the study of the solar-terrestrial relationship. In theory, it is known that solar flares, proton events, and Coronal Mass Ejections can cause sudden and global violent disturbances in the stratosphere, and that atmospheric waves may be stimulated by short-term solar storms(Hood, 1987; Brasseur, 1993; Shindell et al., 2001; Pudovkin, 2004; Gopalswamy et al., 2006; Labitzke, 2006; Thomas et al., 2007; Gray et al., 2010; Shi et al., 2018). However, due to the lack of *in situ*, high-resolution, and continuous observational data in the stratosphere, it is impossible to accurately describe how solar activities affect the mid-latitude stratosphere. Thus, the campaign of Stratospheric Environmental respoNses to Solar stORms (SENSOR), focusing on the above scientific research problem, has been developed (Hu, 2018). SENSOR employs a long-duration zero-pressure balloon as the main platform to carry multiple types of payloads for conducting a series of flight experiments in the mid-latitude stratosphere from 2019 to 2022. These experiments take place during the ascending phase of solar activity. Using a high-altitude balloon as the platform has the advantages of higher temporal and spatial resolution compared to remote sensing, while enabling long-term continuous *in situ* detection. The use of a balloon also allows detection on large horizontal scales during float flight, which cannot be achieved by ground-based equipment.

**After revision:**
The response of the stratosphere to solar activities is an important scientific problem in the study of the solar-terrestrial relationship. In theory, it is known that solar flares, proton events, and Coronal Mass Ejections can cause sudden and global violent disturbances in the stratosphere, and that atmospheric waves may be stimulated by short-term solar storms(Hood, 1987; Brasseur, 1993; Shindell et al., 2001; Pudovkin, 2004; Gopalswamy et al., 2006; Labitzke, 2006; Thomas et al., 2007; Gray et al., 2010; Shi et al., 2018). However, due to the lack of high-resolution, and continuous observational data in the stratosphere, it is impossible to accurately describe how solar activities affect the mid-latitude stratosphere. The campaign of Stratospheric Environmental respoNses to Solar stORms (SENSOR), focusing on the above scientific research, has been developed (Hu, 2018). SENSOR employs a long-duration zero-pressure balloon as the main platform to carry multiple types of payloads for conducting a series of flight experiments in the mid-latitude stratosphere from 2019 to 2022. These experiments take place during the ascending phase of solar activity.

*4- Introduction line 56ff: You explicitly mention that the anemometer should be designed for small- -scale observations in the stratosphere, please specify the requirements for the new device.*

**Thank you very much for your comment. The requirements for the new device have been supplemented.**

In order to measure small-scale fluctuations and also can operate at the float altitude of the balloon, the developed anemometer should achieve the following requirements.

| Table 1 The requirements for the developed anemometer | |
|---|---|
| response time | ≤1s |
| pressure | 1000hPa~30hPa |
| temperature | 30°C ~-70°C |

*5- Introduction line 65ff: You mention the extreme environment in which the device should work: please specify.*

**Thank you for your suggestion. We added the extreme environment our anemometer faced as follows.**

The major challenges for applying an acoustic anemometer in the stratosphere, where has significant difference from terrestrial environment, are the low temperatures with extremes approaching ~-70°C and the low pressures of 30hPa at the height where the high-altitude balloon we used is floating.

*6- Introduction line 68ff: you cite the work of Ovarlez et al., 1978: Why is this ultrasonic anemometer not suitable for your application? Or what want you doing better compared to their device?*

Thank you for your comment. Ovarlez et al. (1978) used transducers resonated at the frequency of 100kHz, according to the analysis of acoustic signals aborption attenuation in the atmosphere (see answer to Q9), the acoustic signals propagation attenuation increases with increasing frequency in the stratosphere. Here we choose transducers at the resonant frequency of 40kHz to obtain a higher SNR relative to 100kHz ones.

*7- Section 2.1: The introduction provides a lot of material you can read in most textbooks about ultrasonic anemometers, the comment on line 106 about the measurement of wind speed in a moving reference system is a general challenge with airborne wind measurements. Please provide at least one citation. However, with a more or less tracer- - like floating balloon this is an even more interesting problem because the balloon motion is quasi- -Lagrangian- this should be discussed in much more detail as it introduce an interesting point.*

Thank you for your suggestion. We added one citation on line 106. And the balloon motion was discussed in answer to Q18.

*8- Section 2.1: There is no discussion at this point how vg is measured on the balloon.*

**Thank you for your comment. We added the explaination how vg is measured on the balloon. Please kindly find the following revision.**

**Before revision:**
Thus, the absolute wind speed, denoted as $v_r$, is the sum of the speed obtained by the anemometer and the speed of the gondola's motion ($v_G$).

**After revision:**

Thus, the absolute wind speed, denoted as $v_r$, is the sum of the speed obtained by the anemometer and the speed of the gondola's motion ($v_G$), measured by the Global Navigation Satellite System(GNSS) installed on the gondola.

*9- Section 2.2 line 119 ff: Here you mention the first time which kind of technical problems and challenges you have with the anemometer under low pressure and temperature conditions. I think this is the key motivation for developing such a system – otherwise you could use a device off the shell – right? This information should be – with much more details – presented at a more prominent place in the manuscript!*

Thank you for your suggestion. We added the following discussion on what efforts we had taken to accommodate our acoustic anemometer to the high-altitude atmosphere in our revised manuscript.

In our experiment, the high-altitude balloon were drifting at the altitude of about 25km, where the atmosphere had significant difference from terrestrial environment: low pressure of about 30hPa and low temperature with extremes approaching ~-70℃ during the balloon's ascent. In order to make sure our anemometer can operate under such an extreme environment, the characteristics of acoustic signals propagation attenuation in the atmosphere had been analyzed.

According to *Bass et al., 1990,* Bass et al., 1995 and *Sutherland and Bass, 2004*, when the sound wave propagates in the atmosphere, the signal attenuation caused by atmospheric absorption is mainly related to the acoustic frequency and atmospheric pressure, attenuation coefficients $\alpha$ in dB per meters (dB/m) can be expressed as follows:

$$\alpha = 8.686f^2\{1.84 \times 10^{-11}\left(\frac{p}{p_0}\right)^{-1}\left(\frac{T}{T_0}\right)^{1/2} + \left(\frac{T}{T_0}\right)^{-5/2} \times \left[0.01278\frac{e^{-2239.1/T}}{f_{r,o}+f^2/f_{r,o}} + 0.1068\frac{e^{-3352/T}}{f_{r,N}+f^2/f_{r,N}}\right]\}$$

Where $f$ is the acoustic frequency in Hz, $p$ is the atmospheric pressure in Pa, $p_0$ is the reference atmospheric pressure in Pa, $T$ is the atmospheric temperature in K, $T_0 = 293.15K$, is the reference atmospheric temperature, $f_{r,o}$, $f_{r,N}$ are the relaxation frequency of molecular oxygen and the relaxation frequency of molecular nitrogen, respectively:

$$f_{r,o} = \frac{p}{p_0}\left(24 + 4.04 \times 10^4 h\frac{0.02+h}{0.391+h}\right),$$

$$f_{r,N} = \frac{p}{p_0}\left(\frac{T_0}{T}\right)^{1/2}\left(9 + 280h\ exp\{-4.17\left[\frac{T_0}{T}^{1/3} - 1\right]\}\right),$$

Where h is the molar concentration of water vapor in percent.

According to the above formulas, figure 5 shows the variation of different frequencies of acoustic signals attenuation caused by atmospheric absorption with height at a distance of 0.2m from the acoustic source.

[Figure]

Figure 5 Atmospheric absorption attenuation of different frequencies of acoustic signals.

The atmospheric absorption attenuation of acoustic signal increases with the increase of acoustic frequency and with the decrease of atmospheric pressure that goes down exponentially with height. At the balloon level flight altitude of about 25km, the received signal intensity with acoustic frequency of 40kHz is at least 10dB higher than that of signals with frequencies of above 100kHz. Therefore, in our acoustic anemometer, the sensors with resonant frequency of 40kHz had been used to achieve higher Signal-to-Noise Ratio (SNR), which is the primary difference between the anemometer that we developed and the anemometers used by Banfield et al. 2016 and Maruca et al. 2017. Besides, an Automatic Gain Control (AGC) circuit is also used, different from terrestrial anemometers, to adjust its gain levels with altitude range to obtain better SNR.

In addition, to avoid the flow distortion from the gondola, the sensor bracket was not mounted outside of the gondola directly, but was through a boom with a length of 1.8m and an elevation angle of 45°. According to Lenoir et al., 2011, the perturbation from the gondola has little influence on the measurements.

*10- Line 137: here you mention the first time typical temperature conditions for your device – I suggest putting it much earlier.*

Thank you for your suggestion. We put them in the introduction part.

*11- Section 2.2 includes in general a lot of technical – and partly trivial - - details that apply for all sensors such as the information that you used a fuse to protect you system. I am more interested in aspects like what is different to other acoustic anemometers and how did you solved problems related to the extreme environment of the stratosphere.*

Please kindly see answers to Q9.

*12- Section 2.3 "Data processing": Similar comment as above: the information you have provided here applies to all ultrasonic anemometers and is generally valid, but not specifically for your device - - at least I don't see any information that refers to the problems caused by low temperature or pressure.*

Thank you for your comment. The following are what we have done to solve problems related to the extreme environment of the stratosphere.

From the detailed discussion in answers to Q9, low atmospheric pressure is the main reason to determine whether an acoustic anemometer can be used in high-altitude environment, since acoustic signals absorption attenuation in the atmosphere will be increased with height. All the efforts we made are to obtain the received signal to noise ratio (SNR) as high as possible, so we have taken improvement measures such as reducing the frequency of acoustic signals and designing AGC circuits. Those are different to other acoustic anemometers and can ensure the anemometer accommodate to the high-altitude atmosphere.

As to data processing, the internal data sampling rate of the acoustic anemometer we developed is 10Hz. In order to further improve the SNR, the original sampled signals within 1s are accumulated in data processing, so the data update rate we given in the manuscript is 1Hz.

*13- Section 2.4: Ground Experiment: The argumentation that both sensors experience the same airflow is somewhat vague; I think one can learn something about the sensors but here it is the first time you mention the temporal resolution of you device which I consider as remarkable low for an ultrasonic anemometer, however, the "reference" system is even slower with a resolution of 10 s? Usually, ultrasoncis are considered as turbulence sensors with sampling frequencies of up to 100 Hz – what are your technical limitations? It might be that your device is slower to meet the conditions of low pressure but you should explain it?*

Thank you for your comment. In ground experiment, we used a small weather sensor (WS500-UMB, Lufft Inc., Germany) as the "reference" system. The sensor's manual shows that the internal sampling frequency is 15Hz and the time interval of instantaneous values can be 1s or 10s (default). The measurements with time interval of 10s had been used in comparison. With regard to our device, as explained in answers to Q12, the original data sampling rate is 10Hz. We accumulated the original sampled signals to improve the received signal-to-noise ratio under low pressure conditions and gave measurements with an update rate of 1Hz.

*14- A technical question: In Fig 5 it seems that the mechanical structure is quite robust but have you considered effects such as flow distortions or transducer shadowing effects which are a key issue for sonics? – Maybe with a lower resolution this is not an issue but you should at least consider such hazards.*

Thank you for your comment. To avoid the flow distortion from the gondola, a long boom with the length of 1.8m and an elevation angle of 45° had been used to keep the sensor bracket at a distance from the gondola. As a result, according to Lenoir et al., 2011, the perturbation from the gondola has little influence on the measurements. In order to minimize the influence of transducer shadowing effects, the distance between sensors should be as far as possible, but this will reduce the signal-to-noise ratio of received signals, or even fail to receive signals at level flight altitude. Therefore, a distance of 0.2m between transducers is a compromise, and we ignored the influence of transducer shadowing effects.

*15- Section 2.4 line 216ff: is the observed wind speed the range you expect as relative airspeed for your balloon experiment? Why do you provide four digits for mean wind speeds? What is the useful resolution (absolute accuracy) of the both sonics?*

As the high-altitude balloon is drifting with the airflow, the relative wind speed, measured by our device, should be very small. We consider the comparison results can meet the wind measurement needs on the balloon platform for most of the time.

I corrected the following sentence according to your suggestion, since the resolutions of our device and

the commercial one are 4cm s$^{-1}$ and 10cm s$^{-1}$, respectively.

**Before revision:**
The mean wind speed difference is −0.0157 m s$^{-1}$, with a standard deviation of 0.3016 m s$^{-1}$.
**After revision:**
The mean wind speed difference is −0.02 m s$^{-1}$, with a standard deviation of 0.30 m s$^{-1}$.

*16- Section 2.4: I am not sure what I can learn from this intercomparison experiment but this is partly also due to the fact that I don't know what you expect from your new device.*

The purpose of intercomparison experiment is to verify the reliability of our device. As there is no available wind tunnel, we choose to carry out the comparative test with a commercial anemometer on the top of the mountain.

*17- Section 3 "Results": I have serious problems with interpreting the wind measurements at all – despite any technical problems such as the observed spikes: I assume even a zero- - pressure balloon with a mass of 400 kg (gondola) does not exactly moves with the mean flow but what are the differences? You measure a relative flow speed but how should I interpret the red line in Fig 7b? By the way, here you only show the zonal wind vector component but you mentioned earlier that you have developed a three- - dimensional wind measurement device – right? What about the other components and how is the 3d- - motion of the balloon be measured? Are you determine also the angular rates and attitude angles?*

**Thank you for your comment.**

We think the gondola does move with the mean flow, and the acoustic anemometer can sense rapid changes in wind and measure the wind speed relative to the gondola. Therefore, the instantaneous absolute wind speed is the sum of the speed obtained by the anemometer and the speed of the gondola's motion. As for figure 7b in the manuscript, we're sorry that we made a mistake. The red line represents the absolute wind speed, not the wind speed relative to the platform. We have corrected this in the revised version.

The zonal wind speed, meridional wind speed and vertical wind speed can be found in our answer to Q1.

The acoustic anemometer is mounted on the gondola, then the relative position and angle relationships between the device and the gondola are determined. The speed and attitude (include the angular rates and attitude angles) of the gondola are continuously monitored by the GNSS and Inertial Navigation Sensor (INS) installed on the gondola. With these data and the angle relationship between the anemometer and the gondola, the zonal wind, meridional wind and vertical wind can be obtained.

*18- If you have a radiosonde available for comparison: why not compare wind speeds? From my point of view, this would be the most obvious thing to do and one could discuss the topic of the quasi- - Lagrangian observation in more detail.*

Thank you for your suggestion.
While the radiosonde data we used are obtained from Golmud Observation Station (94.90°E, 36.42°N), which is about 150km away from Da chaidan District (95.37°E, 37.74°N), and the radiosonde was launched at 12:00 UTC, approximately four hours before the experiment, we didn't compare wind speeds with the radiosonde. Instead, figure 1(b) and figure 2(b) in answer to Q1 compared wind speeds measured by our anemometer with the results from GNSS installed on the gondola, since the gondola travels with the background winds.

A drifting balloon measurement is a quasi-Lagrangian type measurement, which approximately follows the motion of an air parcel. At float altitudes, the balloon oscillates about several hundred meters around 25km due to the balloon's neutral buoyancy oscillation, gravity waves, turbulence, and also the ballast system used to regulate the fly altitude of the balloon. Thus, the balloon flight altitude is not on an isentropic plane, but is considered to be on a constant density surface (equivalent to the density of ~ 0.041kg/m$^3$), that is, the balloon motion is not ideally Lagrangian type motion, but can be considered to be quasi-Lagrangian type motion. A more detailed description about the quasi-Lagrangian observation will be added in the revised version.

**References:**

Bass, H., Sutherland, L. and Zuckerwar, A.: Atmospheric absorption of sound: Update, The Journal of the Acoustical Society of America, **88**(4), 2019-2021, 1990.

Bass, H.E., Sutherland, L.C., Zuckerwar, A.J., Blackstock, D.T. and Hester, D.: Atmospheric absorption of sound: Further developments, The Journal of the Acoustical Society of America, **97**(1), 680-683, 1995.

Kolmogorov, A.N.: The local structure of turbulence in incompressible viscous fluid for very large Reynolds numbers, Cr Acad. Sci. URSS, **30**, 301-305, 1941.

Lenoir, B., Banfield, D. and Caughey, D.A.: Accommodation Study for an Anemometer on a Martian Lander, Journal of Atmospheric and Oceanic Technology, **28**(2), 210-218, DOI: 10.1175/2010jtecha1490.1, 2011.

Sutherland, L.C. and Bass, H.E.: Atmospheric absorption in the atmosphere up to 160 km, The Journal of the Acoustical Society of America, **115**(3), 1012-1032, DOI: 10.1121/1.1631937, 2004.